# Prevalence and Associated Factors for Periodontal Disease among Type I and II Diabetes Mellitus Patients: A Cross-Sectional Study

**DOI:** 10.3390/healthcare11060796

**Published:** 2023-03-08

**Authors:** Manea Musa Alahmari, Hassan M. AlShaiban, Syed E. Mahmood

**Affiliations:** 1Department of Periodontics and Community Dental Sciences, College of Dentistry, King Khalid University, Abha 62529, Saudi Arabia; 2Saudi Board of Preventive Medicine Program, Aseer Directorate of Health Affairs, Ministry of Health, Abha 62523, Saudi Arabia; 3Family and Community Medicine Department, College of Medicine, King Khalid University, Abha 62529, Saudi Arabia

**Keywords:** prevalence, periodontitis, risk factors, type I and II diabetes mellitus, Saudi Arabia

## Abstract

In Saudi Arabia, the prevalence of diabetes mellitus (DM) is high. DM is a risk factor for periodontal disease. The current study aimed to estimate the prevalence and potential associated factors for periodontitis among type I and II DM patients in Abha, Saudi Arabia. A cross-sectional study was conducted among patients with DM who attended the Periodontal Consultant Center at King Khalid University and Aseer Central Hospital, in Abha city, from January 2020 to January 2022. A questionnaire was used to collect relevant data. Patients were assessed for the severity of periodontitis. A total of 499 DM patients were enrolled in this study. The prevalence of periodontitis was 7.4% among type I DM and 46.4% among type II DM patients. The prevalence of periodontitis was significantly associated with age among type I DM patients (*p*-value = 0.001) and type II DM patients (*p*-value = 0.001), respectively, and smoking among type I DM patients (*p*-value = 0.002) and among type II DM patients (*p*-value = 0.000), respectively. Age and smoking were the potential associated factors for the prevalence of periodontitis among type I and II DM. The study provides evidence about the prevalence of periodontitis among DM patients, creates awareness regarding the factors that potentially contribute to worsening periodontal tissues, and the importance of early diagnosis and prevention to avoid the irreversible destruction of the periodontal tissues.

## 1. Introduction

Diabetes mellitus (DM) is a metabolic disorder affecting biochemical and cellular processes within the body of diabetic patients, and it leads to decreased immunity, exhibiting increased susceptibility to infection [1]. A total of 7 million individuals develop diabetes yearly, and by the year 2030, it is expected that 366 million individuals will have the disease in the world [2].

DM is a risk factor for periodontal disease [3]. Periodontitis is characterized by microbial-associated, host-mediated inflammation that ends in periodontal attachment loss [4]. It deteriorates the blood glucose control. The Gram-negative bacterial infections of periodontitis reduce the insulin-mediated glucose uptake by skeletal muscle and reduces the insulin resistance of the body. However, management of periodontal disease will reduce glycated hemoglobin (HbA1c) and improve the DM [5].

Globally, many studies have estimated the prevalence and risk factors for periodontitis among DM patients [1,2,6]. It was reported that there is a significant association between the prevalence of periodontitis among DM and age [7]. Other studies showed that periodontitis prevalence among DM is associated with sex [1,6,8]. Similarly, other studies revealed a significant association between smoking and the prevalence of periodontitis [8,9,10,11]. Other studies [7,10,12] reported the association between HbA1c level and prevalence of periodontitis.

In the Kingdom of Saudi Arabia (KSA), the prevalence of DM among Saudi adults is 23.7% (a total of 26.2% among males and 21.5% among females, respectively) [13]. Data related to periodontal disease among the Saudi population are poor. A study showed that the prevalence of the periodontal disease among the Saudi people in the Najran region is 39% [14]. A prior survey conducted in Abha showed that the prevalence of periodontitis was 36.8%, and the prevalence of mild, moderate, and severe periodontitis was 57.4%, 36.6%, and 4.95%, respectively [15].

There is a need for further studies to be conducted in the region to provide evidence about the prevalence of periodontitis among DM patients and the factors that contribute to the worsening periodontal tissues in these patients. Therefore, the study aimed to estimate the prevalence and potential associated factors for periodontitis among type I and II DM patients in Abha.

## 2. Materials and Methods

### 2.1. Study Design and Population

A cross-sectional study was conducted among DM patients who attended the Periodontal Consultant Center at King Khalid University and Aseer Central Hospital, in Abha city, from January 2020 to January 2022. The study population consisted of the known type 1 or 2 diabetic patients attending the outpatient departments of the above-mentioned health facilities.

### 2.2. Inclusion and Exclusion Criteria

The study enrolled male and female patients who have been diagnosed with type I or II DM at any age. However, the DM patients who had complications, systemic disease other than DM, and underwent periodontal treatment during the past six months were excluded from this study. Lactating or pregnant women were excluded. Those who did not agree to participate in this study were excluded.

### 2.3. Data Collection and Measurements

A questionnaire was used to collect data related to the following variables: age, sex, smoking status (smoking and non-smoking), DM type, HbA1c level, periodontal status (presence or absence of periodontal disease), the severity of periodontal disease (periodontitis stages), and periodontitis grade. Based on answers at the time of data collection, a smoker was defined as a person who has been smoking every day or occasionally and a non-smoker was a person who had never smoked.

### 2.4. Sample Size and Sampling Technique

Optimal sampling size was calculated on the basis of prior prevalence rate of periodontitis of 32% [16]. The sample size was calculated by the formula 4PQ/L 2, where P is the prevalence, Q is 100-P, and L is the absolute precision, i.e., 5%. Approximate minimum sample size came out to be 348. However, 499 individuals were included. Convenient sampling technique was used.

### 2.5. Diabetes Mellitus Diagnosis

In addition, blood chemical analysis for HbA1c level was tested for DM patients; patients who used insulin or hypoglycaemic medicines were diagnosed with DM. Based on the blood chemical analysis of HbA1c level diabetes, control of diabetes was classified into good glycemic control at HbA1c ≤ 7%, moderately glycemic control at HbA1c > 7.01 and ≤ 8.5%, and poorly glycemic control at HbA1c > 8.5% [17].

### 2.6. Periodontitis Diagnosis

The participant was considered to have a healthy periodontal status if the participant had zero clinical attachment loss (CAL) and was considered to have periodontitis if CAL ≥ 1 mm [4]. The severity of periodontitis was classified into four stages based on the degree of CAL. Stage I (early/mild periodontitis) if CAL is between 1.0 –and 2.0 mm. Stage II (moderate periodontitis) if CAL is between 3.0 –and 4.0 mm. Stage III (severe periodontitis) if CAL ≥ 5.0 mm (1, 4). Moreover, the periodontitis patients were classified into three grades based on the estimated future risk of periodontitis progression and responsiveness to standard therapeutic principles [4]. In the clinical context, a patient is considered a periodontitis case if interproximal CAL is detectable at two or more non-adjacent teeth and when (1) (2) buccal or oral CAL ≥3 mm with pocketing >3 mm is detectable at two or more teeth, but the observed CAL cannot be due to non-periodontitis related causes such as 8 a. gingival recession of traumatic origin, b. dental caries extending in the cervical area of the tooth, c. presence of CAL on the distal aspect of a second molar and associated with malposition or extraction of a third molar, d. endodontic lesion draining through the marginal periodontium, and e. occurrence of a vertical root fracture.

Stage one 1–2 mm;

Stage two 3–4 mm;

Stage three equal or more than 5;

Stage four the same with 3 but advanced form.

The following grading in periodontitis diagnosis was used [18]:

Grade A: slow rate of progression, Grade B: Moderate rate of progression, Grade C: rapid rate of progression.

All calibrations were examined by the principal investigator himself.

### 2.7. Data Analysis

The data were coded, entered, and analysed by SPSS (Statistical Package for Social Sciences) version 26.0 version (SPSS Inc., Chicago, IL, USA). According to the Kolmogorov–Smirnov test, the age and HbA1c levels were not in a normal distribution (*p*-value < 0.05). Therefore, the age and HbA1c levels were presented as the median and interquartile range (IQR) at quartile 1 and quartile 3. The age was categorized by the median into ≥37 years and <37 years. The HbA1c levels are categorized by the cut-off point (7%) into good glycemic control (≤7%) and poor glycemic control (>7%). For statistical analysis, a chi-square test was used. The prevalence ratio (PR) with a 95% confidence interval (CI) was calculated. *p*-value < 0.05 was considered statistically significant. Multivariate analysis (multinomial regression model) was further used. Regression analysis was performed using ‘Periodontal disease’ as the dependent variable and the select variables as independent variables.

### 2.8. Ethics Approval and Consent to Participate

Ethical clearance (IRB/KKUCOD/ETH/2021-22/053) was obtained from the Research and Ethics Committee of King Khalid University in Abha city. The aim of the study was explained to all participants. It was carried out following the Declaration of Helsinki. Informed consent was taken from each participant. Assent from the juvenile patients was obtained. Confidentiality of data was assured and ensured.

## 3. Results

### 3.1. Characteristics of the Study Participants

A total of 499 DM patients were included in the present study. Table 1 shows the characteristics of the patients. Out of them, 268 (53.7%) were aged between 21 and 40 years of age, with a median (IQR) age of 37.0 (28.0 and 50.0) years. Nearly 343 (68.7%) DM patients were female, and only 84 (16.8%) were smokers. The majority, 351 (70.3%) DM patients, had type II DM. Nearly 244 (48.9%) of DM patients had good glycemic control, with a median (IQR) HbA1c level of 7.2% (6.4% and 8.0%). Almost 174 DM patients had periodontitis.

The percentage of type I and II DM patients between the age of 21 and 40 years were 78.4% and 43.3%, respectively, and the median (IQR) age of type I and II DM patients was 25.0 (21.0 and 28.0) years and 44.0 (36.0 and 55.0) years, respectively.

The females among type I and II DM patients were 72.2% and 67.2%, respectively. The prevalence of smokers was only 6.1% among type I DM and 21.4% among type II DM patients. The percentage of type I and II DM patients who had good, moderate, and poor glycemic controls were (27.0%, 43.9%, and 29.1%) and (58.1%, 35.6%, and 6.3%), respectively.

### 3.2. Prevalence of Periodontitis among DM Patients

The prevalence of periodontitis was 7.4% among type I DM patients and 46.4% among type II DM patients. The prevalence of mild, moderate, and severe periodontitis among type I and II DM patients were (36.4%, 9.1%, and 54.5%) and (9.2%, 23.9%, and 66.9%), respectively (Table 1). Moreover, the overall prevalence of periodontitis among DM patients was 34.9%. The overall prevalence of severe and grade A periodontitis was 66.2% and 60.3%, respectively.

### 3.3. Potential Associated Factors for Periodontitis among Type I Diabetes Mellitus Patients

Table 2 shows the associated factors for periodontitis among type I DM patients. The prevalence of periodontitis among those aged ≥ 37 years was 100.0%, while the prevalence of periodontitis among those aged < 37 years was 6.8%. However, the prevalence of periodontitis was significantly higher among those aged ≥ 37 years than those aged < 37 years (PR = 14.7, 95% CI:8.1–26.7, and *p*-value = 0.001). Moreover, the prevalence of periodontitis was significantly higher among smokers than non-smokers (33.3% versus 5.8%, PR = 8.2, 95% CI: 1.7–38.9, and *p*-value = 0.002). However, the results show that age and smoking were potential factors associated with periodontitis among type I DM.

Regarding the severity of periodontitis among type I DM patients, the results show that there is not a significant factor associated with the severity of periodontitis among type I DM (*p*-value > 0.05)

### 3.4. Potential Associated Factors for Periodontitis among Type II Diabetes Mellitus Patients

Table 3 shows the associated factors for periodontitis among type II DM patients. The prevalence of periodontitis among those aged ≥ 37 years was 15.3%, while the prevalence of periodontitis among those aged < 37 years was 23.5%. However, the prevalence of periodontitis was significantly higher among those aged ≥ 37 years than those aged < 37 years (PR = 4.0, 95% CI:2.4–6.9, and *p*-value = 0.001). Moreover, the prevalence of periodontitis was significantly higher among smokers than non-smokers (78.7% versus 37.7%, PR = 2.1, 95% CI: 1.7–2.5, and *p*-value = 0.000). However, the results show that age and smoking were potential factors associated with periodontitis in type II DM.

Concerning the severity of periodontitis among type II DM patients, the prevalence of mild, moderate, and severe periodontitis among those aged ≥ 37 compared to those aged < 37 years was (7.9%, 20.0%, and 72.1% versus 17.4%, 47.8%, and 34.8%), respectively. Moreover, the severity of periodontitis was significantly higher among ages ≥ 37 years than < 37 years (χ2 = 12.5 and *p*-value = 0.002). However, the results show that age was a potential factor for periodontitis among type II DM.

Although the prevalence of severe periodontitis was higher among males, smokers, and poor glycemic control as compared to females, non-smokers, and proper glycemic management of type II DM (76.8%, 72.9%, and 75.0 versus 61.7%, 63.5%, and 59.8), there is not statistical significance between the severity of periodontitis among type II DM and sex, smoking, and HbA1c level (*p*-value > 0.05).

As shown in Table 4, type I DM and age up to 37 years was found to be a significant predictor for periodontal diseases in the study sample.

## 4. Discussion

The study aimed to estimate the prevalence of periodontal diseases and associated factors among type I and II DM patients. It showed that type II DM was more prevalent than type I DM, and periodontitis was more prevalent among type II DM than type I DM patients. Age and smoking are potential associated factors for periodontitis prevalence among type I and II DM patients.

Smoking prevalence was 6.1% among type I DM patients and 21.4% among type II DM patients. Regarding HbA1c level, most type I DM patients had moderate glycemic control, while type II DM patients had reasonable glycemic control.

### 4.1. Prevalence of Periodontitis among DM Patients

This study indicates that the prevalence of periodontitis was 7.4% among type I DM patients compared to 46.4% among type II DM patients. Moreover, the overall prevalence of periodontitis among type I and II DM patients was 34.9%. Most periodontitis patients had severe periodontitis in type I and II DM patients. It may be attributable to the effect of DM in damaging the periodontium [16]. Our result agrees with studies in the Najran province of KSA [14] and Hungary [16]. However, our result disagrees with other studies; the prevalence of periodontitis among DM patients is low in a survey carried out in India (25.3%) [2] and high in a study conducted in Korea (43.7%) [7]. Additionally, other studies reported that the prevalence of periodontitis among type II DM patients was high in India (95.1%, 84.5%) (3, 5) and Iraq (95.9%) [6]. The variation might be attributable to the difference in the definition and methods used to diagnose periodontitis, the ages of enrolled participants, and sampling design methods.

### 4.2. Potential Associated Factors for Periodontitis among Type I Diabetes Mellitus Patients

Our results indicated that the prevalence of periodontitis among type I DM patients is 12.7 times more likely to occur in the age group ≥ 37 years than in the age group < 37 years. However, there is a significant association between the prevalence of periodontitis among type I DM and age. Our finding is dissimilar to studies that included participants aged ≥ 30 years in India [2] and Korea [7]. Age itself is not a risk factor, but age-related disorders may facilitate the microbial–inflammatory dysregulation [19]. 

This study found that there is not a significant association between the prevalence of periodontitis among type I DM and sex. Our finding is dissimilar to studies in India [2] and Korea [7]. This discrepancy might be attributable to the characteristics of our study’s participants (all ages are included, and there are more females than males).

This study revealed a significant association between the prevalence of periodontitis among type I DM and smoking. The prevalence of periodontitis is 2.7 times more likely to occur among smokers than non-smokers. However, smoking is an associated factor for the prevalence of periodontitis, which may be because of the systemic and local effects of smoking on periodontal tissues. Our result agrees with the results of studies in India [2], Iraq [6], Korea [7], Serbia [9], Iran [11], and Hungary [17]. However, it disagrees with other studies in Lithuania [10].

Our study indicated that there is not a significant association between the prevalence of periodontitis and the HbA1c level of type I DM. This finding agrees with studies in Korea [7] and Lithuania [10].

Regarding the severity of periodontitis among type I DM patients, the results showed that there is not a significant association between the severity of periodontitis among type I DM patients and age, sex, smoking, and HbA1c level. Our result agrees with a study in Hungary [1], which found no significant association between HbA1c level and the severity of periodontitis.

### 4.3. Potential Associated Factors for Periodontitis among Type II Diabetes Mellitus Patients

Our results indicated that the prevalence of periodontitis among type II DM patients is 4.0 times more likely to occur in the age group ≥ 37 years than in the age group < 37 years. However, there is a significant association between the prevalence of periodontitis among type II DM and age. Our finding is similar to the result of a study in India [5] and dissimilar to other studies, which included participants aged ≥ 30 years in India [1,2], Iraq [6], and Korea [7]. It might be attributable to the nature of our study, which includes participants of all ages.

This study found that there is not a significant association between the prevalence of periodontitis among type II DM and sex. It might be attributable to the fact that there are more females than males. Our finding is similar to the results of studies in India [1], Iraq [6], and Nepal [8] and dissimilar to others in India [2,5] and Korea [7].

This study revealed a significant association between the prevalence of periodontitis among type II DM and smoking. The prevalence of periodontitis is 2.1 times more likely to occur among smokers than non-smokers. However, smoking is an associated factor for the prevalence of periodontitis. It may be explained by the systemic and local effects of smoking on periodontal tissues. Our result agrees with the results of two studies in India [2,9], along with others in Iraq [6], Korea [7], Nepal [8], Lithuania [10], Iran [11], Hungary [17], Serbia [20], and Saudi Arabia [21].

Although the prevalence of periodontitis among type II DM patients is higher among those with poor glycemic control than those with reasonable glycemic control, there is not a significant association between the prevalence of periodontitis and HbA1c level. This finding agrees with studies in Saudi Arabia [20], Italy [21], India [1], and Lithuania [10]. However, it disagrees with the results of two studies in India [3,5]. The variation might be attributable to the difference in the definition and methods used to diagnose periodontitis and HbA1c estimation.

Regarding the severity of periodontitis among type II DM patients, our findings indicated a significant association between the severity of periodontitis among type II DM patients and age. Our finding is similar to studies in Lithuania [10] and Bangladesh [12].

Although severe periodontitis is increased among males, smokers, and those with poor glycemic control as compared to females, non-smokers, and those with proper glycemic management of type II DM, there is not a significant association between the severity of periodontitis among type II DM and sex, smoking, and HbA1c level. Our result agrees with a study in Bangladesh [12], which showed that sex and smoking are not associated factors for the severity of periodontitis. However, our result disagrees with two studies in India [13] and Bangladesh [14], which found a significant association between HbA1c level and the severity of periodontitis. The variation might be attributable to the difference in the ages of enrolled participants and sampling methods. Thus, periodontitis is an early sign of diabetes mellitus and may therefore serve as a valuable risk indicator. Both dentists and physicians need to be aware of the strong connection between periodontitis and T2DM. Controlling these two diseases might help prevent each other’s incidence [22]. The mechanisms that link diabetes and periodontitis are not completely understood, but involve aspects of inflammation, immune functioning, neutrophil activity, and cytokine biology [23]. Both type I and type II diabetes are associated with elevated levels of systemic markers of inflammation [24]. Diabetes increases inflammation in periodontal tissues, with higher levels of inflammatory mediators such as interleukin-1β (IL-1β) and tumour necrosis factor-α (TNF-α) [25,26]. Periodontal disease has been associated with higher levels of inflammatory mediators such as TNF-α in people with diabetes [27]. Accumulation of reactive oxygen species, oxidative stress, and interactions between advanced glycation end products (AGEs) in the periodontal tissues and their receptor (RAGE, the receptor for advanced glycation end products) all contribute to increased inflammation in the periodontal tissues in people with diabetes [25].

For patients with both type II diabetes and periodontitis, nonsurgical periodontal treatment and periodontal maintenance may help to control HbA1c levels [28]. A dental office that treats patients with periodontitis is a suitable location for screening for diabetes by a simple finger stick and validated HbA1c dry spot analysis [29]. An emerging role for dental professionals is envisaged, in which diabetes screening tools could be used to identify patients at high risk of diabetes, to enable them to seek further investigation and assessment from medical healthcare providers [30]. The patients with diabetes mellitus should be informed about their higher risk of developing periodontal diseases [31].

Our study had some limitations. Firstly, a convenience sample is possibly the most important limitation of this study, which prevented some of the variables from being significant. Secondly, a non-random selection of the study participants is also a considerable limitation that might affect the representation of the participants. The status of plaque control, genetic predisposition, the duration of disease, and amount of smoking based on pack per day are important factors which are not considered in the survey. We hope in the future to have all the required resources to conduct multicentric/nationwide studies and include all the important factors in the analysis. However, the data which included diabetic males and females of all ages and the extensive analysis are the strengths of our study.

## 5. Conclusions

The study concludes that the overall prevalence of periodontitis among DM patients was 34.9%, ranging from 7.4% among type I DM patients to 46.4% among type II DM patients. Moreover, age and smoking were the potential factors associated with periodontitis prevalence among type I and II DM. Age was the only possible factor related to periodontitis severity among type II DM patients. Therefore, an analytic study design using a comparison group, e.g., case-control, is recommended to identify the risk factors associated with periodontitis. Our study provides evidence about the prevalence of periodontitis among DM patients and creates awareness regarding the factors that potentially contribute to worsening periodontal tissues. Moreover, it gives information to DM patients about the importance of early diagnosis and prevention to avoid the irreversible destruction of the periodontal tissues.

## Figures and Tables

**Table 1 healthcare-11-00796-t001:** Characteristics of type I and II diabetes mellitus patients.

Characteristics	Type I No. (%)	Type II No. (%)	Total No. (%)
Age group (years):			
≤20	32 (21.6)	0 (0)	32 (6.4)
21–40	116 (78.4)	152 (43.3)	268 (53.7)
41–60	0 (0)	143 (40.7)	143 (28.7)
≥61	0 (0)	56 (16.0)	56 (11.2)
Median (IQR)	25.0 (21.0 and 28.0)	44.0 (36.0 and 55.0)	37.0 (28.0 and 50.0)
Sex:			
Male	41 (27.7)	115 (32.8)	156 (31.3)
Female	107 (72.3)	236 (67.2)	343 (68.7)
Smoking status:			
Smoker	9 (6.1)	75 (21.4)	84 (16.8)
Non-smoker	139 (93.9)	276 (78.6)	415 (83.2)
HbA1c level:			
Good glycemic control (≤7%)	40 (27.0)	204 (58.1)	244 (48.9)
Moderate glycemic control (7.01 and ≤ 8.5%)	65 (43.9)	125 (35.6)	190 (38.1)
Poor glycemic control (>8.5%)	43 (29.1)	22 (6.3)	65 (13.0)
Median (IQR) (%)	8.0 (7.0 and 8.7)	7.0 (6.1 and 7.6)	7.2 (6.4 and 8.0)
Periodontitis:			
Yes	11 (7.4)	163 (46.4)	174 (34.9)
No	137 (92.6)	188 (53.6)	325 (65.1)
Periodontitis stages: (no. = 174)			
Stage I: early/mild periodontitis	4 (36.4)	15 (9.2)	19 (10.9)
Stage II: moderate periodontitis	1 (9.1)	39 (23.9)	40 (23.0)
Stage III: severe periodontitis	6 (54.5)	109 (66.9)	115 (66.1)
Periodontitis grades: (no. = 174)			
Grade A	5 (45.5)	48 (29.4)	53 (30.5)
Grade B	5 (45.5)	100 (61.4)	105 (60.3)
Grade C	1 (9.0)	15 (9.2)	16 (9.2)
Total	148 (29.7)	351 (70.3)	499 (100)

**Table 2 healthcare-11-00796-t002:** The associated factors for periodontitis and its severity among type I diabetes mellitus.

Characteristics	Periodontitis (N = 148)					Severity (N = 11)				
	Yes (N = 11) n (%)	No (N = 137) n (%)	PR ^a^ (95% CI)	χ2	*p* Value	Mild (N = 4) n (%)	Moderate (N = 1) n (%)	Severe (N = 6) n (%)	χ2	*p* Value
Age group (years)										
≥37 years	1 (100.0)	0 (0.0)	14.7 (8.1–26.7)	12.5	0.001	0 (0.0)	0 (0.0)	1 (100.0)	0.9	0.632
<37 years	10 (6.8)	137 (93.2)				4 (40.0)	1 (10.0)	5 (50.0)		
Sex										
Male	3 (7.3)	38 (92.7)	1.0 (0.3–3.5)	0.0	0.974	1 (33.3)	0 (0.0)	2 (66.7)	0.5	0.780
Female	8 (7.5)	99 (92.5)				3 (37.5)	1 (12.5)	4 (50.0)		
Smoking status										
Smoker	3 (33.3)	6 (66.7)	8.2 (1.7–38.9)	9.3	0.002	1 (33.3)	0 (0.0)	2 (66.7)	0.5	0.780
Non-smoker	8 (5.8)	131 (94.2)				3 (37.5)	1 (12.5)	4 (50.0)		
HbA1c level:										
Good glycemic control (≤7%)	5 (12.5)	35 (87.5)	2.4 (0.7–8.5)	2.1	0.153	0 (0.0)	1 (20.0)	4 (80.0)	5.6	0.60
Poor glycemic control (>7%)	6 (5.6)	102 (94.4)				4 (66.7)	0 (0.0)	2 (33.3)		

^a^ prevalence ratio.

**Table 3 healthcare-11-00796-t003:** The associated factors for periodontitis among type II diabetes mellitus.

Characteristics	Periodontitis (N = 351)					Severity (N = 163)				
	Yes (N = 163) n (%)	No (N = 188) n (%)	PR ^a^ (95% CI)	χ2	*p* Value	Mild (N = 15) n (%)	Moderate (N = 39) n (%)	Severe (N = 109) n (%)	χ2	*p* Value
Age group (years)										
≥37 years	140 (55.3)	113 (44.7)	4.0 (2.4–6.9)	28.8	0.001	11 (7.9)	28 (20.0)	101 (72.1)	12.5	0.002
<37 years	23 (23.5)	75 (76.5)				4 (17.4)	11 (47.8)	8 (34.8)		
Sex										
Male	56 (48.7)	59 (51.3)	1.1 (0.7–1.8)	0.4	0554	5 (8.9)	8 (14.3)	43 (76.8)	4.6	0.102
Female	107 (45.3)	129 (54.7)				10 (9.3)	31 (29.0)	66 (61.7)		
Smoking status										
Smoker	59 (78.7)	16 (21.3)	2.1 (1.7–2.5)	39.8	0.001	4 (6.8)	12 (20.3)	43 (72.9)	1.6	0.452
Non-smoker	104 (37.7)	172 (62.3)				11 (10.6)	27 (26.0)	66 (63.5)		
HbA1c level:										
Good glycemic control (≤7%)	87 (42.6)	117 (57.4)	0.8 (0.7–1.0)	2.8	0.093	11 (12.6)	24 (27.6)	52 (59.8)	4.9	0.088
Poor glycemic control (>7%)	76 (51.7)	71 (48.3)				4 (5.3)	15 (19.7)	57 (75.0)		

^a^ prevalence ratio.

**Table 4 healthcare-11-00796-t004:** Predictors of periodontal diseases using multivariate logistic regression analysis.

Variables	OR	*p*-Value	95% Confidence Interval	
			Lower Bound	Upper Bound
DM Type I	13.815	0.000	1.996	9.334
DM Type II	Ref.			
Good glycemic control (≤7%)	0.304	0.582	0.737	1.724
Poor glycemic control (>7%)	Ref			
Age < 37 years	27.083	0.000	2.320	6.417
Age ≥ 37 years	Ref			

## Data Availability

The study data are available from the corresponding author upon reasonable request.

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
