# Peer review of "Prevalence and Associated Factors for Periodontal Disease among Type I and II Diabetes Mellitus Patients: A Cross-Sectional Study"

_healthcare, 2023, doi:10.3390/healthcare11060796_

Round 1

Reviewer 1 Report (Previous Reviewer 4)

Review report:

 I would like to acknowledge the authors for their valuable work on ‘Prevalence and associated factors for periodontal disease among type I and II diabetes mellitus patients: A cross-sectional study’. Though it might be a new topic in Saudi population specially in the Abha region, similar topic has been published earlier in different international journal. No new information has been observed in the result of this manuscript. I would recommend publishing this manuscript in the local journals. I am mentioning some issue which should be corrected before publishing in any journal.

 Abstract:

Line 13: The study…..change this to either ‘the current study….’ Or ‘this study…..’

## Authors did not correct this.

Line 15: DM patients… change to ‘patients with DM…’

## Authors did not correct this.

Line 19-21: kindly rephrase the sentence ‘the prevalence…..’

## Authors did not correct this.

Materials and methods:

Authors mentioned in the revised version that they have calculated the sample size which was 348. Then why additional 151 individuals exposed in this research?

 Results:

1.     It was mentioned in the statistical section that samples were not normally distributed; therefore, median and IQR was used. Why there is mean and SD mentioned in the table 1 for age and HbA1c level? Only median and IQR is enough.

## there are still mean and sd present in the table 1

  References:

Most of the references are not consistent as per the healthcare journal guideline. Kindly make sure all the reference should follow the exact guideline for the references.

## references are still inconsistent.

Author Response

Dear Reviewer, Thank you for your valuable suggestions and helping us to improve the manuscript. Please find a point to point reply.

Reviewer 2 Report (Previous Reviewer 3)

The authors have made the suggested corrections.

Author Response

Dear Reviewer, Thank you for your valuable suggestions and helping us to improve the manuscript.

Reviewer 3 Report (Previous Reviewer 1)

You corrected what I suggested.

Author Response

Dear Reviewer, Thank you for your valuable suggestions and helping us to improve the manuscript. 

Round 2

Reviewer 1 Report (Previous Reviewer 4)

ok

Author Response

Thank you

This manuscript is a resubmission of an earlier submission. The following is a list of the peer review reports and author responses from that submission.

Round 1

Reviewer 1 Report

Dear Authors,

This is an article on an interesting topic. However, some issues need to be addressed before considering it for publication:

Materials and Methods

- More information on the study design and sample could be provided. Which test did you use for sample size calculation?

- More information about the number of examiners and examiner calibration.

- It is better to use multinomial  logistic  regression  analysis to   analyze   the   factors   associated   with periodontal  diseases

- Write the number of Ethics approval

 Discusion

You mention a study limitation, but what is the strength of your study?

Reference

Please write the references according to the Instructions for the authors.

Reviewer 2 Report

The study design is not appropriate, there are many  important factor that can affect periodontal diseases which are not defined and control in this study. 

The amount of smoking based on pocket per day is not considered.

The status of plaque control, genetic predisposition, the duration of disease are important factors which are not considered.

The mechanism of effect of diabetes on periodontal diseases are not discussed in discussion.

The ethict code is not in the article.

Reviewer 3 Report

This manuscript was written to assess the prevalence and the potential associated factors for periodontitis among type I and II Diabetes Mellitus (DM) patients in Abha, Saudi Arabia. The results of this study are essential for dental researchers and professionals. This article highlights the prevalence of periodontitis among patients suffering from types I and II DM, as well as its association with age and smoking. However, the manuscript requires significant grammatical modifications. Some sentences having relevant information were difficult to understand because of the way they were drafted. The authors need to review the whole manuscript carefully and make the necessary changes including, but not limited to, the ones mentioned below:

1.       P1L12 and P1L27: Change to uppercase ‘M’ in ‘mellitus’ (DM)

2.       P1L30: Either use ‘D.M.’ or ‘DM’ for the abbreviation. Do not use both alternatively.

3.       P1L31: ‘decreased’

4.       P1L32: The line ‘It is affecting 245 million individuals in the world.’ Is not specific. What is the reference for this number? When was this number reported?

5.       P1L33: ‘it is expected’.

6.       P1L36: ‘that ends in’

7.       P1L37: Change to ‘It deteriorates the blood glucose control’.

8.       P1L38-39: Change to ‘and reduces the insulin resistance of the body’.

9.       P2L55-56: This sentence is grammatically incorrect. Please modify.

10.   P2L66-68: Change it to: ‘However, the DM patients who had complications, systemic disease other than DM, and underwent periodontal treatment during the past six months were excluded from this study. Lactating or pregnant women were also excluded.’

11.   P2L78: This sentence is grammatically incorrect. Please modify.

12.   P2L84: Change to: ‘The participant was considered to have a healthy periodontal status if the participant had zero clinical attachment loss (CAL), and was considered to have periodontitis if CAL ≥ 1 mm.’

13.   P6L193-195: Please keep the font type and size consistent with the rest of the article.

14.   P7L250-253: Please keep the font type and size consistent with the rest of the article.

15.   P8L262-263: This sentence is grammatically incorrect. Please modify.

Reviewer 4 Report

Review report:

I would like to acknowledge the authors for their valuable work on ‘Prevalence and associated factors for periodontal disease among type I and II diabetes mellitus patients: A cross-sectional study’. Though it might be a new topic in Saudi population specially in the Abha region, similar topic has been published earlier in different international journal. No new information has been observed in the result of this manuscript. I would recommend publishing this manuscript in the local journals. I am mentioning some issue which should be corrected before publishing in any journal.

Abstract:

Line 13: The study…..change this to either ‘the current study….’ Or ‘this study…..’

Line 15: DM patients… change to ‘patients with DM…’

Line 19-21: kindly rephrase the sentence ‘the prevalence…..’

Introduction:

1.      Line 30: Diabetes mellitus shortened as DM in abstract. Should be consistent with the short form either ‘D.M.’ or ‘DM’.

2.      Should be consistent with the spacing. Some sentences have space before the reference number and some places missed spacing.

3.      Line 49: add ‘a total of’ 26.2% among males……

Materials and methods:

1.      Kindly add the manufacturer of the SPSS software.

2.      Line 95: change ‘However’ to ‘therefore’ or ‘hence’.

3.      Line 105: Authors only mentioned about the informed consent. What about the assent form for the juvenile patients?

Results:

1.      Though in the limitation, it was mentioned that the convenient sampling technique was used in this study, need to be mentioned in the method and result section before mentioning the total sample for this study.

2.      It was mentioned in the statistical section that samples were not normally distributed; therefore, median and IQR was used. Why there is mean and SD mentioned in the table 1 for age and HbA1c level? Only median and IQR is enough.

Discussion:

1.      Font size and format should be consistent. i.e Line 193-195 and 250-253.

2.      Authors discussed the topic in wide range; however, the most probable reason for the dis similarity of the outcome of the previous studies were not discussed. The probable reason for the dissimilar outcome comparing to the previous studies should briefly be discussed.

References:

Most of the references are not consistent as per the healthcare journal guideline. Kindly make sure all the reference should follow the exact guideline for the references.